# Molecular mechanism to target the endosomal Mon1-Ccz1 GEF complex to the pre-autophagosomal structure

Jieqiong Gao[1], Lars Langemeyer[1], Daniel Kümmel[2], Fulvio Reggiori[3], Christian Ungermann[1]*

[1]Biochemistry Section, Department of Biology/Chemistry, University of Osnabrück, Osnabrück, Germany; [2]Structural Biology Section, Department of Biology/Chemistry, University of Osnabrück, Osnabrück, Germany; [3]Department of Cell Biology, University Medical Center Groningen, University of Groningen, Groningen, Netherlands

**Abstract** During autophagy, a newly formed double membrane surrounds its cargo to generate the so-called autophagosome, which then fuses with a lysosome after closure. Previous work implicated that endosomal Rab7/Ypt7 associates to autophagosomes prior to their fusion with lysosomes. Here, we unravel how the Mon1-Ccz1 guanosine exchange factor (GEF) acting upstream of Ypt7 is specifically recruited to the pre-autophagosomal structure under starvation conditions. We find that Mon1-Ccz1 directly binds to Atg8, the yeast homolog of the members of the mammalian LC3 protein family. This requires at least one LIR motif in the Ccz1 C-terminus, which is essential for autophagy but not for endosomal transport. In agreement, only wild-type, but not LIR-mutated Mon1-Ccz1 promotes Atg8-dependent activation of Ypt7. Our data reveal how GEF targeting can specify the fate of a newly formed organelle and provide new insights into the regulation of autophagosome-lysosome fusion.

DOI: https://doi.org/10.7554/eLife.31145.001

*For correspondence:
cu@uos.de

Competing interests: The authors declare that no competing interests exist.

## Introduction

Macroautophagy, called here autophagy, is an important quality control pathway, during which cellular material such as organelles and cytosolic components are engulfed by a double-membrane vesicles, the autophagosome (*Shibutani and Yoshimori, 2014*; *Mizushima et al., 2011*). In both yeast and mammals, autophagosome formation is a complex process that begins with the assembly of the phagophore or isolation membrane. Once complete, the autophagosome first fuses with endosomes to form an amphisome and then with lysosomes in mammalian cells, while it directly fuse with the lysosome-like vacuole in yeast (*Lamb et al., 2013*; *Chen and Klionsky, 2011*).

How autophagosomes become fusion competent with lysosomes is still poorly understood. Like for other fusion events, autophagosome fusion with vacuoles or endosomes requires SNAREs, Rab GTPases (Rabs) and the HOPS tethering complex (*Reggiori and Ungermann, 2017*; *Barr, 2013*; *Kümmel and Ungermann, 2014*). Rabs have a central role in this fusion cascade. They are held soluble in the cytosol by the GDP-dissociation inhibitor (GDI) proteins, which bind GDP-loaded Rabs. Once on membranes, a Rab-specific guanine nucleotide exchange factor (GEF) converts Rabs into their active GTP-form (*Barr, 2013*). This allows their interaction with effectors such as tethering factors (*Kümmel and Ungermann, 2014*). The Rab7 GTPase is required for the fusion of endosomes with lysosomes and lysosomal transport (*Nordmann et al., 2012*). In yeast, the Rab7-homolog Ypt7 binds to the HOPS tethering complex in this process, which in turn supports SNARE assembly and fusion. Rab7 as well as Ypt7 are also required for fusion of autophagosomes with endosomes

**eLife digest** Autophagy is a word derived from the Greek for "self-eating". It describes a biological process in which a living cell breaks down its own material to release their chemical building blocks that can then be used to make new molecules. Autophagy is often triggered when a cell becomes damaged or when nutrients are in short supply. The hallmark of autophagy is the formation of structures called autophagosomes. These structures capture the cellular material, fuse with other compartments in the cell – namely endosomes in animals and vacuoles in yeast – and then deliver the material inside, ready to be broken down.

For an autophagosome to fuse to an endosome or a vacuole, small proteins of the Rab protein family must be located on the surface of the autophagosome. Rab proteins are recruited to this surface by enzymes known as GEFs. However it remains unclear how most GEFs get to the surface of a compartment within the cell to begin with.

The Mon1-Ccz1 complex is a GEF that occurs in yeast and animals, including fruit flies and humans. It is found on endosomes, and was recently shown to also localize to autophagosomes. Now, Gao et al. report that, in yeast, the Mon1-Ccz1 complex binds directly to a protein named Atg8. This protein is anchored on to the surface of autophagosomes, and is closely related to other proteins in animal cells.

Gao et al. discovered that this specific GEF binds to Atg8 via at least one binding site on its Ccz1 component. This binding site is only needed for the GEF to localize to the autophagosomes; the Mon1-Ccz1 complex can still bind to endosomes without it. Blocking the GEF from binding to Atg8 stopped the autophagosomes from fusing with vacuoles.

These findings reveal how a GEF can be targeted to two distinct compartments in the cell: endosomes and autophagosomes. Further work is now needed to understand how this process is regulated by the availability of nutrients or damage to the cell, to ensure that autophagy is only triggered under the right conditions. Also, because cancer cells often rely on autophagy to survive, the molecules that regulate this process could represent possible targets for new anticancer drugs.
DOI: https://doi.org/10.7554/eLife.31145.002

(*Gutierrez et al., 2004*; *Ganley et al., 2011*; *McEwan et al., 2015*) and detected on autophagosomes (*Hegedűs et al., 2016*).

The conserved Mon1-Ccz1 GEF complex triggers endosomal maturation by activating Ypt7 (or Rab7 in metazoans) primarily on late endosomes (*Nordmann et al., 2010*; *Gerondopoulos et al., 2012*; *Singh et al., 2014*; *Cui et al., 2014*), but likely also on autophagosomes (*Hegedűs et al., 2016*). In agreement with this notion, it has been shown that yeast Mon1-Ccz1 is essential for autophagy progression (*Wang et al., 2002*). As Mon1-Ccz1 can interact with Rab5-GTP, Rab5 may promote Rab7 recruitment to endosomes, possibly with support by the local generation of phosphatidylinositol-3-phosphate (PI-3-P) (*Singh et al., 2014*; *Hegedűs et al., 2016*; *Cui et al., 2014*). It remains unresolved, however, how Mon1-Ccz1 is specifically targeted to autophagosomes to trigger SNARE-mediated fusion (*Figure 1A*). The SNAREs involved in this event have been implicated in previous studies (*Darsow et al., 1997*; *Fischer von Mollard and Stevens, 1999*; *Dilcher et al., 2001*; *Sato et al., 1998*; *Reggiori and Ungermann, 2017*).

Atg8 is one of 16 conserved autophagy-related (Atg) proteins, which are essential for autophagosome formation, and it possesses six mammalian homologues (*Shpilka et al., 2012*). Members of the Atg8/LC3 protein family are conjugated to phosphatidylethanolamine (PE) at the autophagosome membrane, and interact with several Atg proteins via a LC3 interacting region (LIR motif) to control both maturation and fusion (*Wild et al., 2014*; *Nakatogawa et al., 2007*; *Klionsky and Schulman, 2014*; *Abreu et al., 2017*). Here, we demonstrate that Atg8 recruits the endosomal GEF Mon1-Ccz1 to the pre-autophagosomal structure. Mutants in a LIR motif present in the Ccz1 C-terminal do not impair GEF activity or endosomal function, but block autophagosome fusion with vacuoles. Our data thus reveal how a GEF can mark two different organelles with the same Rab for fusion via distinct mechanisms.

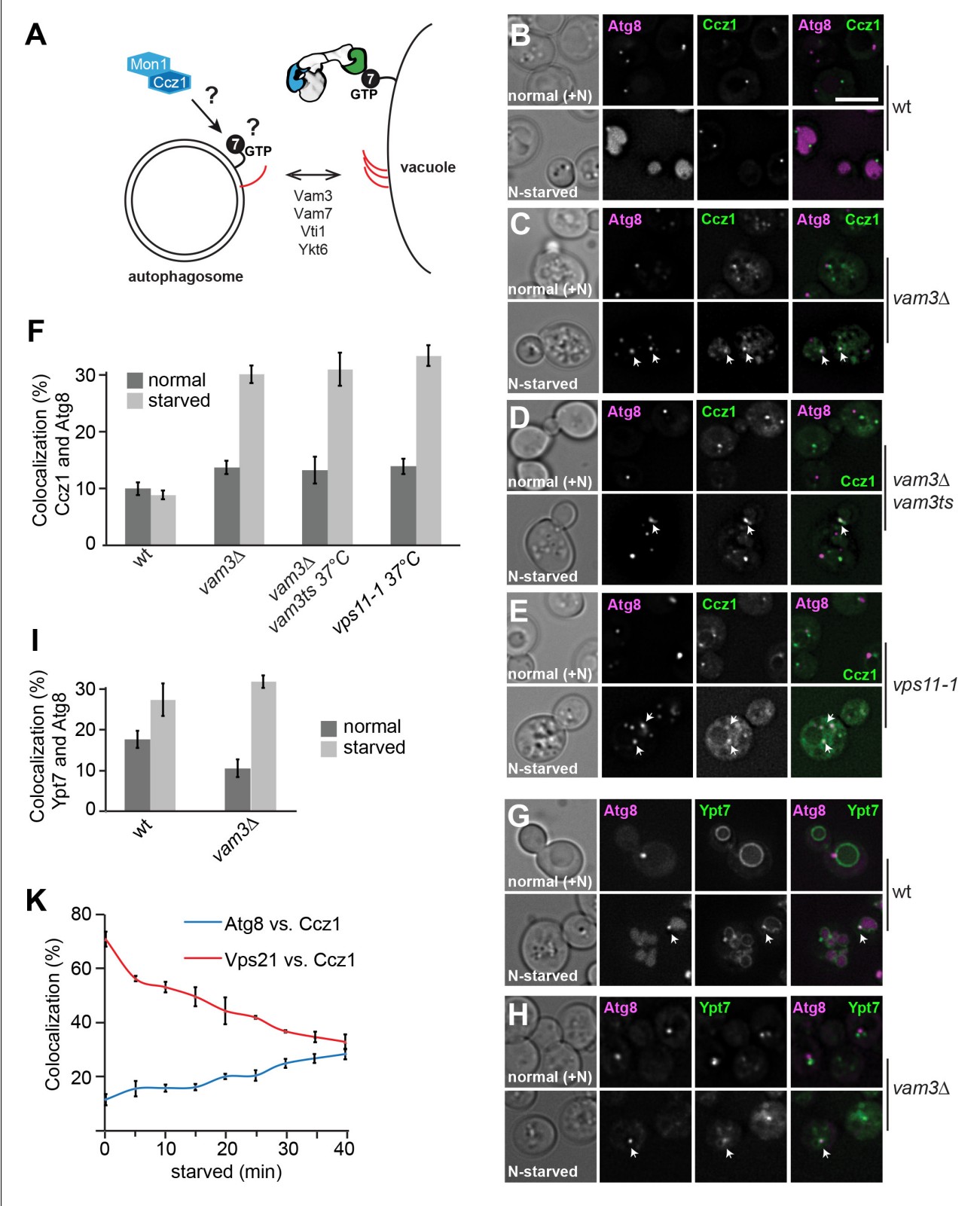

**Figure 1.** Mon1-Ccz1 and Ypt7 localize to autophagosomes during starvation. (**A**) Working model of autophagosome-vacuole fusion. Red lines indicate the involved SNAREs Vam3, Vam7, Vti1, and Ykt6. Ypt7 is shown bound to the HOPS complex. For details see text. (**B–H**) Localization of Atg8 relative to Ccz1 and Ypt7 during growth and nitrogen starvation. Cells expressing mCherry-tagged Atg8 or GFP-tagged Ccz1 or Ypt7 were grown in YPD (normal, +N) or in synthetic medium without nitrogen (SD-N, labeled as N-starved) for 2 hr and analyzed by fluorescence microscopy and showed via

*Figure 1 continued on next page*

*Figure 1 continued*

individual slices. Size bar, 5 μm. (F–I) Percentage of Ccz1 puncta (F) or Ypt7 puncta (I) co-localizing with Atg8 under both conditions. Atg8 dots (n ≥ 50), Ccz1 dots (n ≥ 300) and Ypt7 dots (n ≥ 200) were quantified by Image J. Error bars represent standard deviation (SD). (K) Relocalization of Ccz1 during starvation. Time course of mCherry-tagged Vps21 and Atg8 relative to GFP-tagged Ccz1. Error bars represent SD.

DOI: https://doi.org/10.7554/eLife.31145.003

The following source data and figure supplements are available for figure 1:

**Source data 1.** Quantification of percentage of Ccz1 puncta or Ypt7 puncta co-localizing with Atg8 during growth and nitrogen starvation for *Figure 1F,I*.

DOI: https://doi.org/10.7554/eLife.31145.006

**Source data 2.** Quantification of percentage of Ccz1 puncta co-localizing with Atg8 or Vps21 during growth and nitrogen starvation in different time points for *Figure 1K*.

DOI: https://doi.org/10.7554/eLife.31145.007

**Source data 3.** Quantification of ALP activity for nitrogen starvation 2 hr and 4 hr in wild-type and *vps21Δ* cells.

DOI: https://doi.org/10.7554/eLife.31145.008

**Figure supplement 1.** Mon1 localizes to autophagosomes during starvation.

DOI: https://doi.org/10.7554/eLife.31145.004

**Figure supplement 2.** Deletion of the Rab5 like Vps21 results in autophagy defects.

DOI: https://doi.org/10.7554/eLife.31145.005

## Results

To determine how yeast autophagosomes are specifically decorated with Ypt7, we analyzed the sub-cellular distribution of both Mon1 and Ccz1 as the GEF complex formed by these two proteins (*Nordmann et al., 2010*). In particular, we co-localize GFP-tagged Mon1 and Ccz1 with mCherry-tagged Atg8, an autophagosome marker protein (*Suzuki et al., 2007*), in wild type cells in growing and nitrogen starvation conditions, which induce autophagy. In yeast, autophagosomes form at the pre-autophagosomal assembly site proximal to the ER and vacuole, which is visible as a single dot-like structure by fluorescence microscopy (*Klionsky et al., 2016*; *Graef et al., 2013*; *Suzuki et al., 2013*; *Mari and Reggiori, 2010*). Ccz1 and Mon1 were found in distinct puncta, likely endosomes (*Rana et al., 2015*; *Nordmann et al., 2010*), which were not co-localizing with the Atg8 puncta in nutrient-rich conditions (*Figure 1B*; *Figure 1—figure supplement 1A*). After nitrogen starvation, however, Atg8 labeled the vacuole lumen in wild-type cells as expected (*Rieter et al., 2013*). This made it impossible to localize Ccz1 or Mon1 to autophagosomes under these conditions, because of their rapid fusion with the vacuole upon completion (*Geng et al., 2008*). We therefore employed different strategies to block fusion of autophagosomes with vacuoles to determine whether Ypt7, Ccz1, and Mon1 transiently co-localize with Atg8. Deletion of the vacuolar Qa-SNARE Vam3, or temperature sensitive mutants of either Vam3 or the HOPS subunit Vps11 block fusion processes with the vacuole (*Darsow et al., 1997*; *Peterson and Emr, 2001*). When cells with these mutations were starved, we indeed observed an accumulation of Atg8-positive autophagosomes, and both Ccz1 and Mon1 were markedly co-localizing with them (*Figure 1C–E*, quantified in F; *Figure 1—figure supplement 1B,C*). Likewise, a fraction of Ypt7 colocalized with Atg8 in *vam3Δ* cells only during starvation (*Figure 1G–I*). In agreement with this, purified autophagosomes contained both Ypt7 and Mon1-Ccz1 on their surface (Gao and Ungermann, in preparation). Furthermore, we analyzed GFP-Ypt7 in cells overexpressing Ape1. Ape1 overexpression results in the formation of a giant Ape1 oligomer, which is too large to be closed by the isolation membrane marked by mCherry-Atg8 (*Suzuki et al., 2013*). We found that Ypt7 localizes on the cup-shaped isolation membrane concentrated in a dot in wild-type and *vam3Δ* cells (*Figure 1—figure supplement 1*). These data support our interpretation that Ypt7 is present on the autophagosomal membrane. To determine whether starvation promotes the redistribution of Ccz1 to autophagosomes relative to endosomes, we monitored Ccz1 co-localization with Atg8 or Vps21, an endosomal marker protein (*Cabrera et al., 2013*), over time. Indeed, the fraction of Ccz1 in Vps21-positive organelles decreased, while the localization to Atg8-positive puncta increased during the monitored time period (*Figure 1K*). As recently published (*Zhou et al., 2017*), we found the *vps21Δ* mutant displays impaired autophagy as monitored by the processing of initially cytosolic Pho8Δ60 in the vacuole lumen (*Figure 1—figure supplement 2E*). We also noticed that Ccz1 is cytosolic in *vps21Δ* cells before and after starvation, which did not allow us to detect this protein on autophagosomal structures (*Figure 1—figure supplement 2A–D*).

It is possible that the localization of Mon1-Ccz1 to endosomes is a prerequisite for a later the movement of the GEF complex to autophagosomes during starvation.

These data suggest that the Mon1-Ccz1 complex is specifically recruited to autophagosomes. To monitor the potential contribution of Atg proteins, including Atg8, in targeting Mon1-Ccz1 and Ypt7 to autophagosomes, we selected the precursor Ape1 oligomer (*Kim et al., 1997*), a specific autophagosomal cargo behaving similar to Atg8 under starvation conditions, for a small colocalization screen. In wild-type cells and in agreement with the data obtained using mCherry-Atg8 (*Figure 1B, G*; *Figure 1—figure supplement 1A*), the starvation-induced co-localization with Ape1 was observed for Ypt7 but not for Ccz1 (*Figure 2A,B*; *Figure 2—figure supplements 1* and *2*). To clarify the contribution of known Atg proteins in this process, we generated double knock out mutants lacking *VAM3* and selected *ATG* genes and repeated the assay. In *vam3Δ atg1Δ* cells as in *vam3Δ* cells Ccz1 and Ypt7 both robustly colocalized with Ape1 upon nutrient deprivation (*Figure 2A,B*; *Figure 2—figure supplements 1* and *2*). However, all the mutants blocking Atg8 conjugation to PE such as those lacking the components of the conjugation machinery or Atg8 itself, abolished colocalization of Ccz1 and Ypt7 with Ape1. Interestingly, the deletion of Atg14, a subunit of the PI-3-kinase I complex required for autophagy (*Kihara et al., 2001*), did not impair colocalization of Ccz1 and Ape1 on autophagosomes (*Figure 2A*; *Figure 2—figure supplement 1*), though affected Ypt7 colocalization with Ape1 (*Figure 2B*; *Figure 2—figure supplement 2*). Colocalization of Ape1 relative to Atg8 was not affected in the *atg14* mutant (*Figure 2—figure supplement 2*). This suggests that PI-3-P is not a main determinant for Mon1-Ccz1 targeting to autophagosomes, though might support its activity and/or recruitment of Ypt7.

Taken together these observations indicate that Mon1-Ccz1 recruitment onto autophagosome requires Atg8. To determine whether Mon1-Ccz1 binds to Atg8 in vivo, we immunoprecipitated GFP-tagged Atg8 from wild-type and *atg4Δ* cells co-expressing Ccz1-TAP. Atg4 is required for processing of Atg8 prior to its lipidation on the preautophagosomal structure (*Chen and Klionsky, 2011*). In agreement with our previous finding, we observed an interaction of Ccz1 with Atg8 in wild-type cells, which was greatly enhanced when cells were starved prior to lysis. In contrast, no interaction was observed in *atg4Δ*, supporting our notion that Ccz1 binds to lipidated Atg8 on autophagosomal structures in vivo (*Figure 2C*). We next investigated whether Mon1-Ccz1 could bind to Atg8 directly. We thus incubated purified Mon1-Ccz1 with immobilized GST-Atg8 or ubiquitin, and detected robust binding only to Atg8 (*Figure 2D*). Atg8 recognizes LIR motifs via its N-terminal helices (*Klionsky and Schulman, 2014*). We therefore tested if truncation mutants of Atg8 lacking the 8 or 24 N-terminal residues still bind Mon1-Ccz1. Importantly, binding was now lost strongly suggesting that Mon1-Ccz1 specific association to Atg8 is mediated by one or more LIR motifs (*Figure 2E*). To further test whether this interaction depends on the Ccz1 LIR motif(s), we generated an Atg8 I21R mutant, which blocks the binding pocket for the crucial $W_0$ LIR motif residue (*Noda et al., 2008*). We observed no binding between Atg8 I21R and Ccz1 (*Figure 2E*), indicating that Atg8 indeed recognizes a LIR motif in Ccz1. This Atg8 mutant functions in non-selective autophagy (*Figure 2—figure supplement 3A–C*), yet has some defect in selective autophagy when we followed Ape1 processing during starvation (*Figure 2—figure supplement 3D*). It thus behaves like previously characterized mutants at this site (*Noda et al., 2008*).

We then asked which part of Mon1-Ccz1 binds to Atg8. Mon1 and Ccz1 interact with each other via their conserved longin domains (*Nordmann et al., 2010*; *Cabrera et al., 2014*), which form a common interface that is required for specific Ypt7 activation (*Kiontke et al., 2017*). However, Mon1 has some additional 150 residues at the N-terminus of its longin domain, and both Mon1 and Ccz1 have C-terminal domains, whose structure and function is so far unresolved. We therefore generated N- and C-terminal truncation mutants of both proteins and monitored localization and autophagy. In starvation conditions, GFP-tagged truncation mutants of Mon1 expressed in the *mon1Δ* background did not impair vacuole morphology or starvation-induced Atg8 trafficking to the vacuole lumen (*Figure 2G*). We noted though that GFP-Mon1 localization was more strongly impaired in the N-terminal than the C-terminal truncation. In contrast, deletion of the C-terminal domain of GFP-tagged Ccz1 resulted in fragmented vacuoles, even though Ccz1 was still localized to distinct puncta that did not co-localize with Atg8 (*Figure 2G*). We thus asked whether Ccz1 alone might be able to directly interact with purified Atg8. Although purified Mon1-Ccz1 as well as Ccz1 alone were able to bind GST-Atg8 (*Figure 2D*), a mutant complex of Mon1 with Ccz1ΔC showed

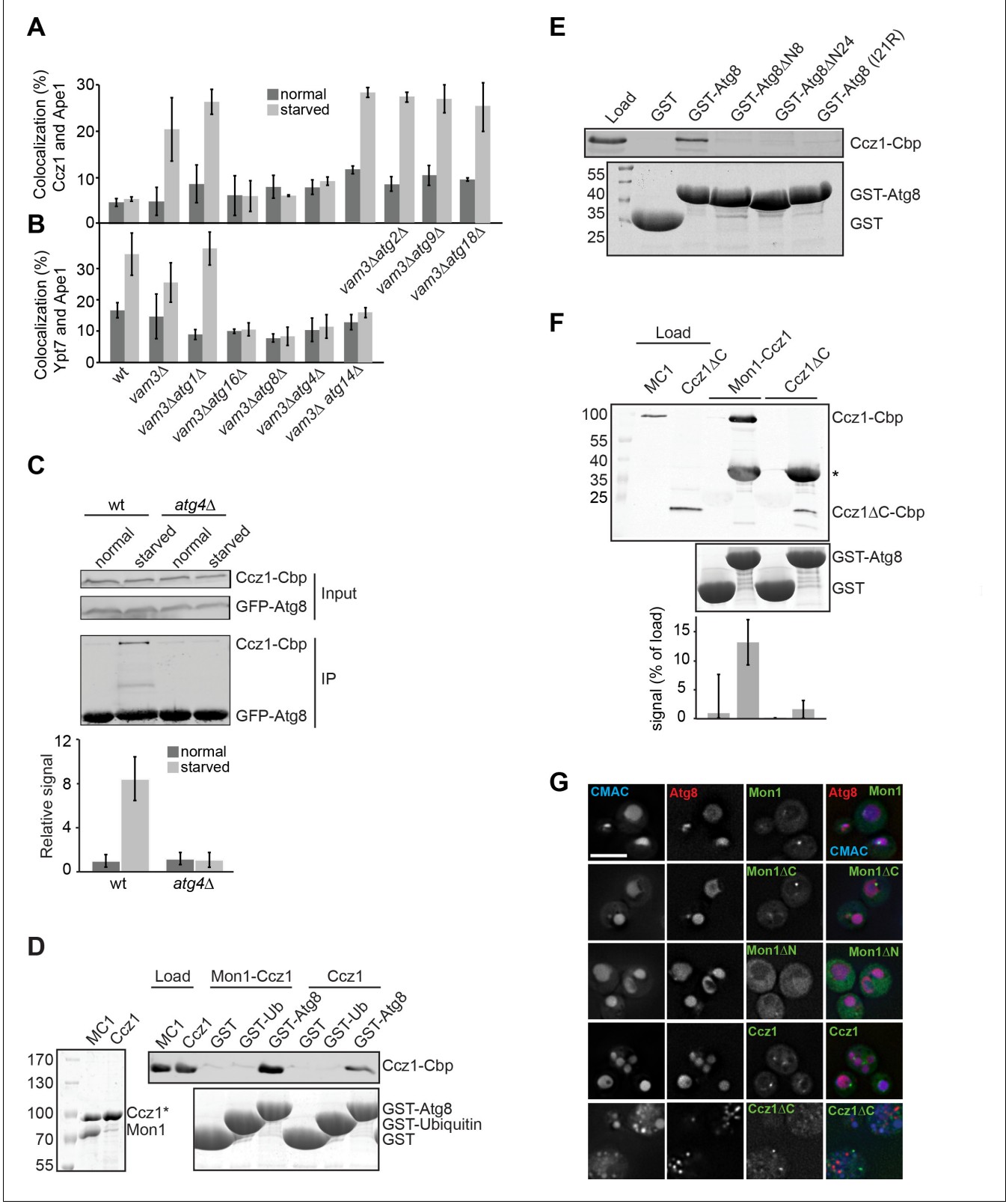

**Figure 2.** Atg8 binds to Mon1-Ccz1 via the Ccz1 C-terminal part. (A–B) Atg8 is required for localization of Ccz1 to autophagosomes. Graphs show percentage of colocalization of Ccz1 puncta (A) or Ypt7 puncta (B) relative to Ape1 puncta in wild-type and the different mutants. Cells were grown and analyzed as in *Figure 1*. Ape1 dots (n ≥ 50), Ccz1 dots (n ≥ 300), and Ypt7 dots (n ≥ 200) were quantified by Image J. Error bars represent SD. (C–E) Interaction analysis of Atg8 with Mon1-Ccz1. (C) Immunoprecipitation of TAP-tagged Ccz1 from wild-type and *atg4Δ* strain co-expressing GFP-Atg8.
*Figure 2 continued on next page*

*Figure 2 continued*

The strain was grown in YPD or in SD-N for 3 hr before preparing cell extracts. GFP-Atg8 was subsequently immunoprecipitated using GFP-trap beads. Finally, immunoprecipitates were analyzed by Western blotting against GFP and CbP-tag. The graph is the quantification of three independent experiments, where the interaction observed in unstarved cells from wild-type is set as 1. Error bars are SD. (D) Interaction of Atg8 with Mon1-Ccz1 or purified Ccz1. TAP-tagged proteins (shown as purified proteins on Coomassie stained gels to left) were incubated with GST, GST-ubiquitin, and GST-Atg8 immobilized on GSH-Sepharose. Eluted proteins were resolved by SDS-PAGE and analyzed by Western blotting against the CbP-tag (top) or by Coomassie staining (bottom). Load, 10%. (E) Interaction of Atg8 mutants with Mon1-Ccz1. Analysis was done as in (D) with the indicated GST-tagged Atg8 truncation mutants. (F) Interaction of Mon1-Ccz1ΔC with Atg8. Mon1-Ccz1ΔC was purified as wild-type and analyzed for interaction with GST-tagged Atg8 as before. Top, Western blot against the CbP tag. A star indicates the additional decoration of GST-Atg8 by the antibody; bottom, Coomassie staining and quantification of three experiments. (G) Requirements of Mon1 and Ccz1 domains for autophagy. The indicated truncations were analyzed in cells expressing mCherry-tagged Atg8. Vacuoles were stained with CMAC, and cells grown in SD-N medium were then analyzed by fluorescence microscopy as in *Figure 1B*. Size bar, 5 μm.

DOI: https://doi.org/10.7554/eLife.31145.009

The following source data and figure supplements are available for figure 2:

**Source data 1.** Quantification of percentage of Ccz1 puncta or Ypt7 puncta co-localizing with Ape1 during growth and nitrogen starvation for *Figure 2A,B*.

DOI: https://doi.org/10.7554/eLife.31145.013

**Source data 2.** Quantification of the interaction between Atg8 and Mon1-Ccz1 during growth and nitrogen starvation from wild-type and *atg4Δ* cells for *Figure 2C*.

DOI: https://doi.org/10.7554/eLife.31145.014

**Source data 3.** Quantification of interaction of Mon1-Ccz1ΔC with Atg8 for *Figure 2F*.

DOI: https://doi.org/10.7554/eLife.31145.015

**Source data 4.** Quantification of ALP activity for nitrogen starvation 3 hr in wild-type and Atg8 I21R mutant cells.

DOI: https://doi.org/10.7554/eLife.31145.016

**Figure supplement 1.** Ccz1 fails to localize to autophagosomes during starvation in mutants that lack Atg8 or the conjugation system.

DOI: https://doi.org/10.7554/eLife.31145.010

**Figure supplement 2.** Analysis of Ypt7 and Atg8 localization relative to Ape1 in wild-type and mutant strains.

DOI: https://doi.org/10.7554/eLife.31145.011

**Figure supplement 3.** The Atg8 I21R mutant shows impaired selective autophagy.

DOI: https://doi.org/10.7554/eLife.31145.012

strongly reduced interaction (*Figure 2F*). Altogether, these observations suggest that the C-terminal part of Ccz1 directs the GEF complex to Atg8-positive autophagosomes.

## Identification of putative LIR motifs in Ccz1

To determine the direct binding site in the Ccz1 C-terminal, we compared the C-termini of multiple Ccz1 homologs. As metazoan Ccz1 is shorter than yeast Ccz1, we narrowed our search on the conserved fragment and identified the putative LIR motifs (https://ilir.warwick.ac.uk; *Figure 3A*). We generated the corresponding mutants by changing the aromatic $W_0$ and the hydrophobic $L_4$ residues into alanines. Among the nine mutants (*Figure 3A*), two showed impaired GFP-Ccz1 localization to mCherry-Atg8-positive autophagosomes under nitrogen starvation conditions, that is Y236A V239A (named *LIR1*) and Y445A L448A (*LIR2*) (*Figure 3B,C,E*). These two motifs are highly conserved across species (*Figure 3B*). However, we noticed that trafficking of mCherry-tagged Atg8 to the vacuole was not totally compromised in the single mutants at normal growth temperature (*Figure 3C*). We therefore combined both mutations and nitrogen starved the cells. This resulted in a complete block of autophagy in the double mutant as shown by defects in Atg8 delivery and processing in the vacuole (*Figure 3C and F*), but also vacuole morphology (*Figures 3C* and *4D*). Under these conditions, numerous mCherry-Atg8-positive autophagosomes accumulated in the cytoplasm, consistent with a defect in fusion with vacuoles. The *LIR1,2* mutant behaves thus as the Ccz1ΔC mutant, and is likewise compromised in both autophagy and vacuole biogenesis in general (*Figures 2F* and *3C,F*).

We therefore focused on the single mutants. We were wondering why the single LIR mutants were still functional, even though Ccz1 targeting seemed diminished. We considered the possibility that the LIR mutants may be impaired at higher temperature, and thus repeated the starvation assay at 37°C (*Figure 3D*). Although the wild-type cells were functional in autophagy, both LIR mutants now accumulated Atg8-positive autophagosomes in cells (*Figure 3D*, quantified in E).

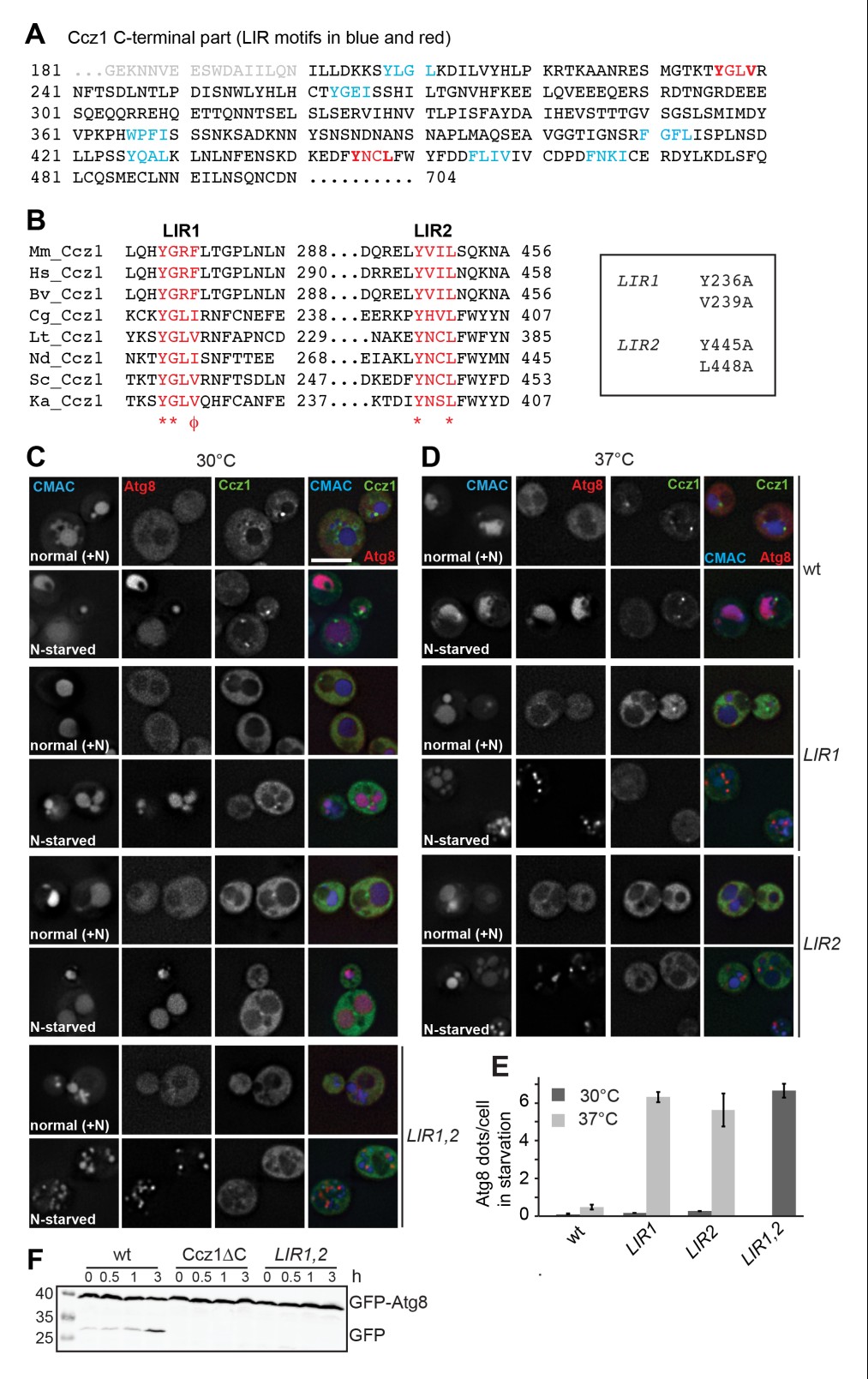

**A** Ccz1 C-terminal part (LIR motifs in blue and red)

```
181 ...GEKNNVE ESWDAIILQN ILLDKKSYLG LKDILVYHLP KRTKAANRES MGTKTYGLVR
241 NFTSDLNTLP DISNWLYHLH CTYGEISSHI LTGNVHFKEE LQVEEEQERS RDTNGRDEEE
301 SQEQQRREHQ ETTQNNTSEL SLSERVIHNV TLPISFAYDA IHEVSTTTGV SGSLSMIMDY
361 VPKPHWPFIS SSNKSADKNN YSNSNDNANS NAPLMAQSEA VGGTIGNSRF GFLISPLNSD
421 LLPSSYQALK LNLNFENSKD KEDFYNCLFW YFDDFLIVIV CDPDFNKICE RDYLKDLSFQ
481 LCQSMECLNN EILNSQNCDN .......... 704
```

**B**

```
              LIR1                        LIR2
Mm_Ccz1  LQHYGRFLTGPLNLN 288...DQRELYVILSQKNA 456
Hs_Ccz1  LQHYGRFLTGPLNLN 290...DRRELYVILNQKNA 458
Bv_Ccz1  LQHYGRFLTGPLNLN 288...DQRELYVILNQKNA 456
Cg_Ccz1  KCKYGLIRNFCNEFE 238...EERKPYHVLFWYYN 407
Lt_Ccz1  YKSYGLVRNFAPNCD 229....NAKEYNCLFWYFN 385
Nd_Ccz1  NKTYGLISNFTTEE  268...EIAKLYNCLFWYMN 445
Sc_Ccz1  TKTYGLVRNFTSDLN 247...DKEDFYNCLFWYFD 453
Ka_Ccz1  TKSYGLVQHFCANFE 237....KTDIYNSLFWYYD 407
             **  ϕ                *   *
```

| LIR1 | Y236A |
| | V239A |
| | |
| LIR2 | Y445A |
| | L448A |

**Figure 3.** Identification of the LIR motifs in Ccz1 required for function. (**A**) Schematic representation of potential LIR motifs of the C-terminal part of Ccz1. Blue and red indicates all LIR motifs analyzed, red the motifs that also impair Ccz1 localization. (**B**) Alignments of the potential Ccz1 LIR motifs Mm: *Mus musculus*, Hs: *Homo sapiens*, Cg: *Candida glabrata*, Lt: *Lachancea thermotolerans*, Nd: *Naumovozyma dairenensis*, Ka: *Kazachstania Africana*. (**C–D**) Effect of LIR mutants on localization, autophagy and vacuole morphology. Analysis was done as in *Figure 1B–H*. CMAC staining was done for 15

*Figure 3 continued on next page*

*Figure 3 continued*

min before analysis. Cells were grown either at 30°C or 37°C during growth or starvation. Size bar, 5 μm. (E) Quantification of Atg8 dots per cell from images in (C–D). Error bars represent SD. (F) Analysis of autophagy over time. Cells were grown at 30°C and incubated in starvation medium for the indicated time periods, then harvested, and proteins were analyzed by SDS-PAGE and Western blotting against GFP.

DOI: https://doi.org/10.7554/eLife.31145.017

The following source data is available for figure 3:

**Source data 1.** Quantification of Atg8 dots per cell from Ccz1 wild-type and LIR mutants for *Figure 3E*.

DOI: https://doi.org/10.7554/eLife.31145.018

To test if these LIR mutants indeed compromise binding to Atg8, we produced and used the mutants in Atg8 binding assays (see *Figure 2D–F*). Both *LIR1* and the *LIR1,2* double mutants could be purified as wild-type Mon1-Ccz1 from yeast, indicating that they were not destabilizing the complex (*Figure 4A*). However, they showed poor interaction with Atg8 (*Figure 4B,C*). As we encounter major problems in the purification of the Mon1-Ccz1 complex with *LIR2*, we did not further pursue it in our in vitro analyses. Nonetheless, these data agree with a model, where one and possibly two Ccz1 LIR motifs are required for the recruitment of Mon1-Ccz1 to Atg8.

## The Ccz1 LIR motifs are not required for endosomal trafficking

Our data suggest an important function of the one and possibly two LIR motifs in directing Mon1-Ccz1 to autophagosomes. As vacuole morphology of the *LIR1* and *LIR2* mutants was only mildly impaired during heat stress (*Figures 3D* and *4D*), we asked if endosomal trafficking was functional in these mutants. The vacuolar hydrolase carboxypeptidase Y (CPY), which is normally sorted from the Golgi via the endosome to the vacuole, is lost from cells in mutants impaired in vacuole biogenesis such as *vps39Δ* or the temperature sensitive mutant *vps11-1* at 37°C (*Figure 4E*). Likewise, *ccz1Δ* cells have less intracellular CPY. However, both LIR mutants in Ccz1 were entirely unperturbed also at elevated temperature or when cells were starved. As a second assay, we traced the sorting of the methionine transporter Mup1 from the plasma membrane to the vacuole (*Arlt et al., 2015*). In both wild-type cells and the LIR mutants, Mup1-GFP was mainly at the plasma membrane in the absence of methionine, but was efficiently sorted to the vacuole lumen when methionine was added after the temperature shift to 37°C (*Figure 4F*). This sorting remained unaffected at higher temperatures as well. We therefore conclude that the *LIR1* and *LIR2* mutants selectively disable Mon1-Ccz1 targeting to autophagosomes, whereas endosomal function of Mon1-Ccz1 remains unperturbed under the same conditions.

## Atg8 specifies Mon1-Ccz1 function on autophagosomal membranes

Our data imply that Atg8 is indeed a primary determinant to recruit Mon1-Ccz1 to autophagosomes. We used our Mon1-Ccz1 *LIR1* mutant to directly test this hypothesis as this was the best behaving complex. From previous in vitro experiments with purified organelles and proteins we have learned that mutations can compromise protein function in vitro much more clearly than in vivo (*Bröcker et al., 2012*; *Ungermann et al., 1999*). We therefore took advantage of GEF assay that we developed before to monitor Mon1-Ccz1 activity on membrane-bound Ypt7 (*Cabrera et al., 2014*). C-terminally His-tagged Ypt7 was preloaded with the MANT-GDP nucleotide, which looses fluorescence when exchanged for non-fluorescent GTP. In the presence of liposomes carrying the His-interacting DOGS-NTA lipid, and the nucleotide exchange reaction is strongly enhanced when Mon1-Ccz1 is also recruited onto the liposome surface (*Cabrera et al., 2014*). Using this assay, we compared wild-type and *LIR1* mutated Mon1-Ccz1 complex (*Figure 5A*). Both complexes had similar activity for Ypt7 (*Figure 5A,D*). We then lowered the Mon1-Ccz1 concentration in our assay to test whether Mon1-Ccz1 targeting and function could depend on membrane-bound Atg8. Indeed, membrane-targeted Atg8-His, but not soluble Atg8, stimulated the GTP exchange reaction (*Figure 5B, D*), presumably due to its ability to recruit the GEF complex to membranes. In contrast, the Mon1-Ccz1 LIR mutant did not respond to the addition of Atg8 (*Figure 5C,D*). Our data thus show that membrane-bound Atg8 can recruit Mon1-Ccz1 to membranes to promote Ypt7 activation.

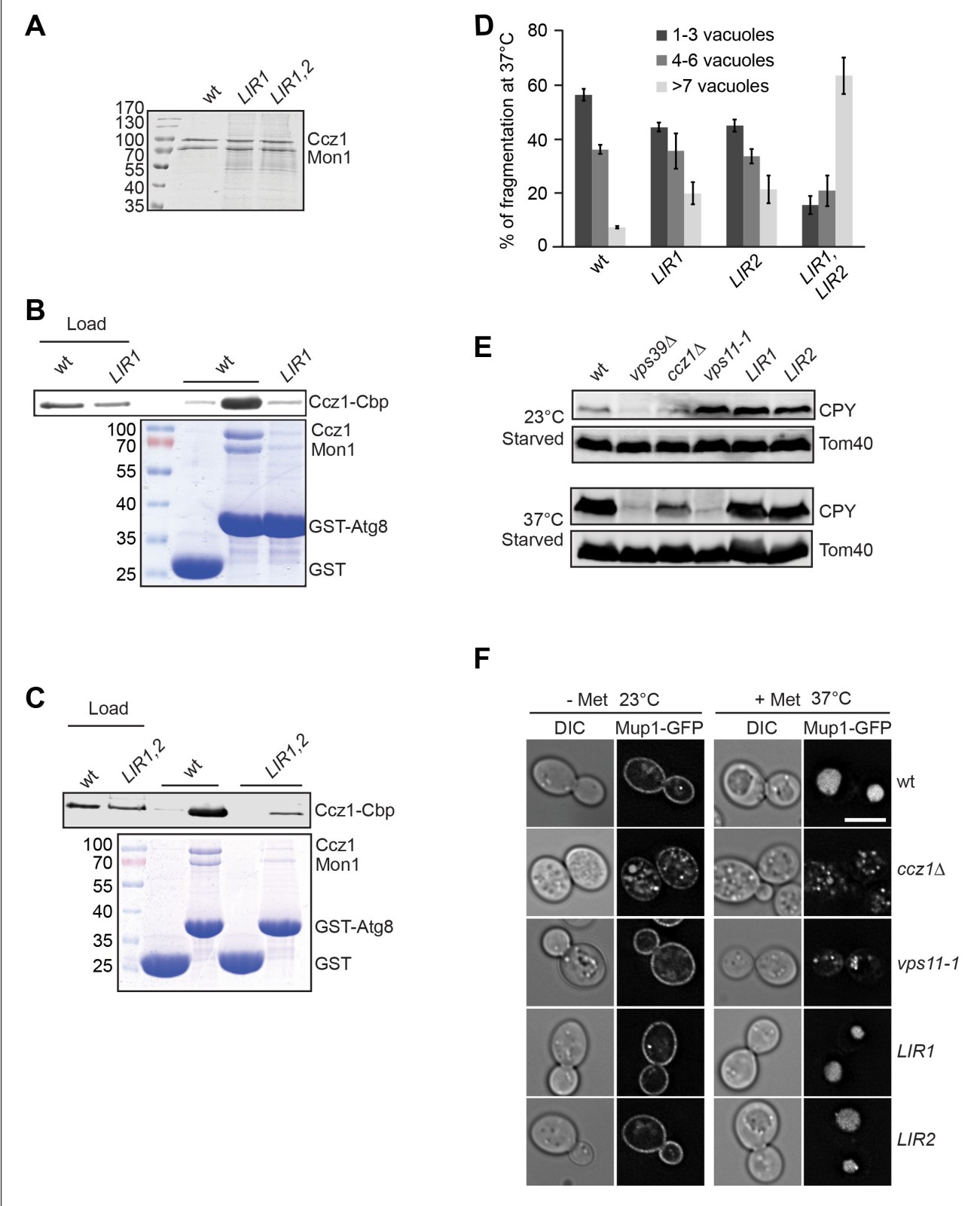

**Figure 4.** LIR motifs in Ccz1 are required for Atg8 binding, but not for the endocytic pathway. (A–C) Interaction of Ccz1 LIR mutants with Atg8. (A) Analysis of purified Mon1-Ccz1 wild-type and mutant complex by SDS-PAGE and Coomassie staining. All of strains were grown at 30°C for purification. (B–C) Mutations in the LIR motifs impair Mon1-Ccz1 interaction with Atg8. Interaction analysis was done as in *Figure 2D*, and proteins were analyzed by Western blotting (top) and Coomassie staining (bottom). (D) Comparison of vacuole morphology in LIR mutant cells. Cells were grown at 30 or 37°C in

*Figure 4 continued on next page*

*Figure 4 continued*

starvation medium, and vacuoles were then stained with CMAC. The number of vacuoles per cell was quantified as indicated. Error bars, SD. (**E**) Effect of *LIR* mutants on sorting of vacuolar hydrolases. The indicated cells were grown in starvation medium at the indicated temperature for 2 hr. Total cell lysates were generated and proteins were resolved on SDS-PAGE. Western blots were decorated against CPY and Tom40 (as loading control). (**F**) Endocytosis analysis in *LIR* mutants. The indicated cells expressing Mup1-GFP were grown in the absence (-Met) of methionine in minimal medium to an $OD_{600}$ of 1.0 at the 23°C. Where indicated, methionine was added after the temperature shift to 37°C, and cells were analyzed by fluorescence microscopy after 1 hr. Size bar, 5 μm.

DOI: https://doi.org/10.7554/eLife.31145.019

The following source data is available for figure 4:

**Source data 1.** Quantification of vacuole morphology in LIR mutant cells for *Figure 4D*.

DOI: https://doi.org/10.7554/eLife.31145.020

# Discussion

Previous studies showed that artificial redirection of GEFs can redirect different Rabs to other membranes (*Blümer et al., 2013*; *Gerondopoulos et al., 2012*), yet the molecular determinants that target GEFs to their correct membrane are only partially known and rarely experimentally validated. Autophagosomes form de novo and finally fuses with lysosomes (*Reggiori and Ungermann, 2017*). Like maturing endosomes, autophagosomes need to acquire the machinery to allow their fusion with lysosomes, including the Rab7/Ypt7 GTPase. Here we have uncovered a simple molecular mechanism that specifically targets the GEF Mon1-Ccz1 onto the surface of autophagosomes. The Ccz1 subunit has at least one conserved C-terminal LIR motif, which directly binds to the LC3 homolog Atg8. Once on autophagosomes, Mon1-Ccz1 recruits and activates the Rab7-like Ypt7 from the cytosol, which in turn can bind the HOPS tethering complex to trigger SNARE-mediated fusion. We indeed found recent evidence that Mon1-Ccz1 is sufficient to activate Ypt7, which was provided in a soluble complex with GDI, and thus triggered fusion in a reconstituted assay (*Langemeyer et al., 2018*). Similarly, the TRAPP GEF complexes could activate their corresponding Rab-GDI complexes on membranes (*Thomas and Fromme, 2016*). In agreement with this interpretation, only wild type but not LIR-mutated Mon1-Ccz1 strongly promotes Ypt7 activation in the presence of membrane-localized Atg8 (*Figure 5E*).

Our data imply that lipidated Atg8 is a specific determinant to redirect Mon1-Ccz1 to autophagosomes. In addition to Atg8, PI-3-P may support re-localization to both endosomes and autophagosomes (*Hegedűs et al., 2016*; *Cabrera et al., 2014*). Indeed, deletion of Atg14 in *Drosophila* fat cells appears to affect autophagosome fusion in addition to altering the biogenesis of these vesicles (*Hegedűs et al., 2016*). The generation of autophagosomal PI-3-P is required for multiple events, including efficient Atg8 lipidation (*Shibutani and Yoshimori, 2014*). Interestingly, Mon1-Ccz1 localization to Atg8 positive dots was not impaired if synthesis of the autophagosome-specific PI-3-P pool was blocked by *atg14* deletion (*Figure 2C*). We consider it therefore unlikely that PI-3-P synthesis is a primary factor for Mon1-Ccz1 localization to autophagosomes. In contrast, our analysis suggests that PI-3-P may be critical for Mon1-Ccz1 activity, which could explain the defect in Ypt7 localization to autophagosomes of the *atg14Δ* mutant. Alternatively, PI-3-P might directly support the recruitment of Ypt7, even though we have evidence that Mon1-Ccz1 activity is most critical in this process (*Langemeyer et al., 2018*). How the reported PI-3-P binding (*Lawrence et al., 2014*; *Cabrera et al., 2014*) affects Mon1-Ccz1 function needs to be further dissected. Future studies will also need to explore how Mon1-Ccz1 is timely and spatially recruited to autophagosomes.

Importantly, our study reveals that Mon1-Ccz1 is functional in the endocytic pathway, when its LIR motifs are singularly mutated. This provides further evidence that Mon1-Ccz1 has a dual role and two different targeting mechanisms for two distinct organelles. By identifying the LIR mutants, we established one of the few conditions that might accumulate fully assembled autophagosomes, which are incompetent of fusing with vacuoles, while maintaining endosome-vacuole fusion and thus vacuoles functional.

In mammalian cells, additional proteins such as PLEKHM1 (*McEwan et al., 2015*) have been identified as factors involved in the fusion between autophagosomes and lysosomes. Interestingly, PLEKHM1 directly binds LC3-like proteins and Rab7, and could thus support HOPS-mediated tethering and fusion of autophagosomes with lysosomes. We believe that GEFs such as Mon1-Ccz1 are

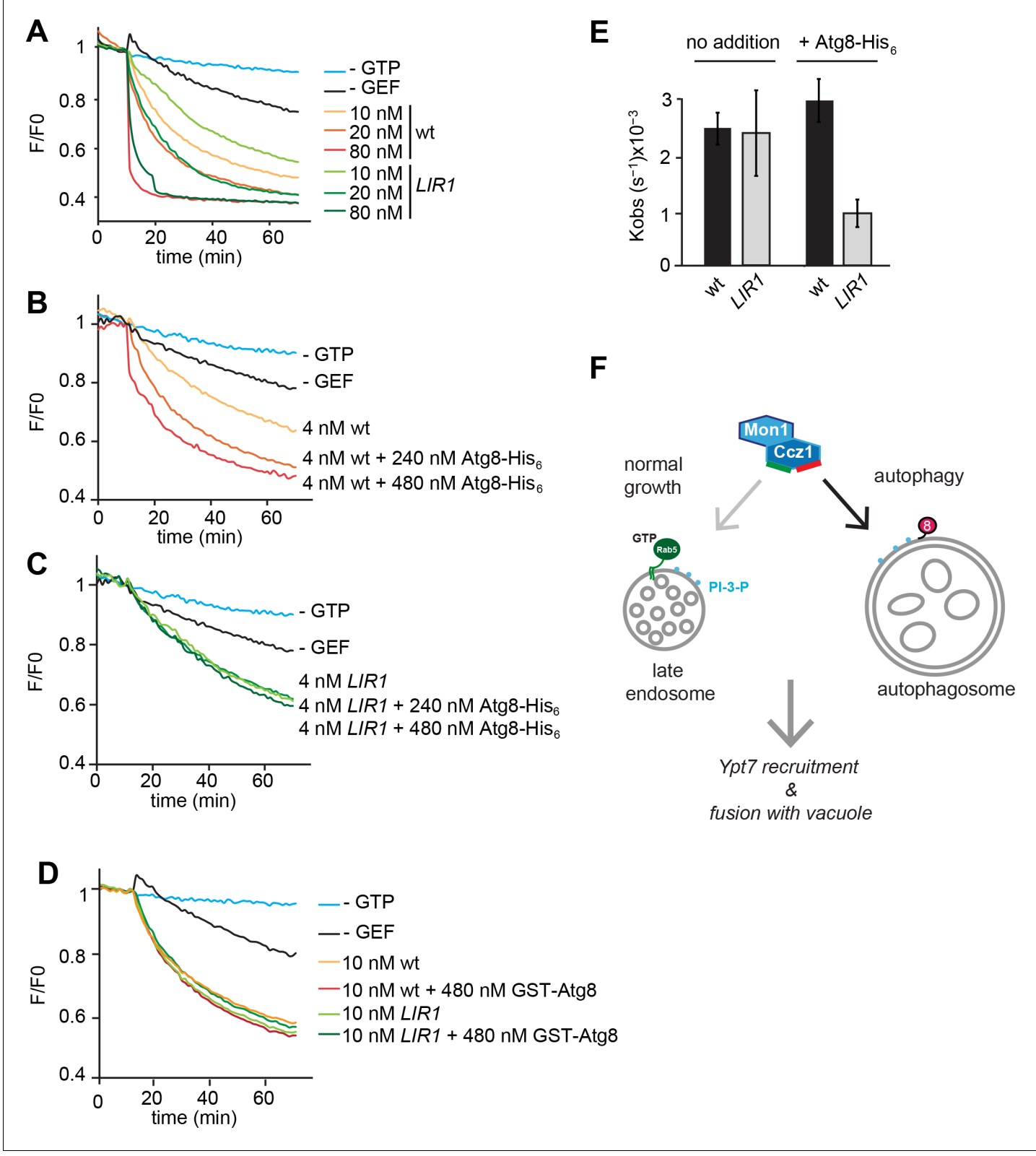

**Figure 5.** Functional reconstitution of Atg8-dependent GEF activity of Mon1-Ccz1. (**A**) GEF activity of wild-type and mutant Mon1-Ccz1 complex. GEF activity was monitored by displacement of MANT-GDP from Ypt7 using a microplate reader (see Materials and methods). Assay was carried out with liposomes capable of binding His-tagged Ypt7 (*Cabrera et al., 2014*). Without GTP, blue line; without GEF, black line; wt refers to different concentrations of Mon1-Ccz1, LIR1 to the Mon1-Ccz1 mutant complex. (**B–D**) Effect of membrane-bound Atg8 or soluble Atg8 on GEF activity. Analysis

*Figure 5 continued on next page*

*Figure 5 continued*

was carried out as in (A) with reduced Mon-Ccz1 concentrations and upon addition of His-tagged Atg8 at the indicated concentrations. (E) Quantification of the rate constants of wild-type and mutant Mon1-Ccz1 complex in the presence and absence of Atg8 for *Figure 5B–C*. Rate constants were calculated based on the initial slope of the GEF curve (n = 3) (*Kiontke et al., 2017*; *Langemeyer et al., 2014*). Error bars, SD. (F) Model of Mon1-Ccz1 recruitment to the autophagosome and endosomes. For details see text.
DOI: https://doi.org/10.7554/eLife.31145.021
The following source data is available for figure 5:

**Source data 1.** GEF activity of wild-type and mutant Mon1-Ccz1 complex for *Figure 5A*.
DOI: https://doi.org/10.7554/eLife.31145.022
**Source data 2.** Effect of membrane-bound Atg8 on GEF activity for *Figure 5B,C*.
DOI: https://doi.org/10.7554/eLife.31145.023
**Source data 3.** Effect of soluble Atg8 on GEF activity for *Figure 5D*.
DOI: https://doi.org/10.7554/eLife.31145.024
**Source data 4.** Quantification of the rate constants of wild-type and mutant Mon1-Ccz1 complex in the presence and absence of Atg8 for *Figure 5E*.
DOI: https://doi.org/10.7554/eLife.31145.025

the most critical factors to confine Rab localization and thus determine organelle identity. The cooperation with LC3-like proteins could then provide a combinatorial code to target GEFs and additional fusion factors to autophagosomes. Interestingly, Atg8 is not homogenously distributed over the surface of forming autophagosomes (*Graef et al., 2013*), and could potentially cluster fusion factors to facilitate their cooperation during fusion. How Atg8 recycling and fusion are then coordinated (*Abreu et al., 2017*), it is yet another exciting riddle to be dissected. At least Mon1-Ccz1 localization to autophagosomes might be dispensable, once Rab7/Ypt7 is recruited and bound to HOPS.

Recent work of us and others revealed that GEFs can recruit Rab GTPases from the GDI complex to membranes (*Langemeyer et al., 2018*)(*Thomas and Fromme, 2016*). The identification of Atg8 as a determinant for Mon1-Ccz1 localization to autophagosomes provides the first example of how a GEF can be diverted to a different location. Differential spatiotemporal recruitment of GEFs allows cells to operate distinct pathways, such as autophagy and endosomal maturation, depending on their metabolic needs while employing the same machinery. For endosomal localization, Rab5-GTP has been suggested as a Mon1-Ccz1 interactor based on yeast-two-hybrid interactions (*Li et al., 2015*; *Cui et al., 2014*; *Singh et al., 2014*; *Kinchen and Ravichandran, 2010*). Future studies will need to dissect if this order of events can be indeed recapitulated in vitro and how further endosomal and autophagosomal factors specify GEF localization.

## Materials and methods

### Yeast strains and molecular biology

Strains and plasmids used in this study are listed in *Supplementary file 1* and *2*, respectively. Deletions and tagging of genes were done by homologous recombination of respective PCR fragments (*Janke et al., 2004*; *Puig et al., 1998*). Mon1 and Ccz1 mutants were generated by QuikChange mutagenesis (Stratagene, La Jolla, CA). Mon1 and Ccz1 truncation mutants have been published (*Kiontke et al., 2017*). Plasmids encoding GST-Atg8 and Atg8-His6 were kindly provided by Ivan Dikic (Goethe University School of Medicine, Frankfurt am Main, Germany), and Sascha Martens (University of Vienna, Austria), respectively.

### Tandem affinity purification

Tandem affinity purification was performed as described (*Bröcker et al., 2012*; *Lürick et al., 2017*). Six liters of culture in YPG were grown at 30°C to $OD_{600}$ of 6, and cells were harvested and lysed in lysis buffer (300 mM NaCl, 50 mM HEPES-NaOH, pH 7.4, 1.5 mM MgCl$_2$, 1 × FY protease inhibitor mix (Serva, Germany), 0.5 mM PMSF and 1 mM DTT). Lysates were centrifuged for 1 hr at 100,000 *g*, and the cleared supernatant was incubated with IgG Sepharose beads (GE Healthcare, Penzberg, Germany) for 2 hr at 4°C. Beads were collected by centrifugation at 800 *g* for 2 min, and washed with ice cold 15 ml lysis buffer containing 0.5 mM DTT and 10% glycerol. Bound

proteins were eluted by TEV cleavage overnight at 4°C. Purified proteins were analyzed on SDS-PAGE.

### E.coli protein expression and purification

Atg8 was purified from *E. coli* BL21 (DE3) Rosetta cells. Cells were grown to an $OD_{600}$ of 0.6 and induced with 0.5 mM IPTG overnight at 16°C. Cells were lysed in lysis buffer (50 mM HEPES/NaOH, pH 7.5, 150 mM NaCl, 1 mM PMSF, 1x protease inhibitor cocktail (1x = 0.1 mg/ml of leupeptin, 1 mM o-phenanthroline, 0.5 mg/ml of pepstatin A, 0.1 mM Pefabloc)). Lysates were centrifuged for 20 min at 30,000 *g*, and the cleared supernatant was incubated with Glutathione Sepharose (GSH) beads (for GST-tagged proteins) or Ni-NTA beads (for His-tagged proteins) for 1 hr at 4°C on a nutator. Beads were washed with 20 ml cold lysis buffer (GSH-beads) or lysis buffer containing 20 mM imidazole (Ni-NTA beads). Bound proteins were eluted with buffer containing 15 mM reduced glutathione (GSH-beads) or buffer containing 300 mM imidazole (Ni-NTA beads). Buffer was exchanged to 50 mM HEPES/NaOH, pH 7.4, 150 mM NaCl, and 10% glycerol by using a NAP-10 column (GE Healthcare, Penzberg, Germany).

## GST pull down binding assays

To perform GST pull down binding assays, GST or GST-fused Atg8 wild type or Atg8 mutants or ubiquitin were used as bait, and Mon1-Ccz1 was used as a prey. GST or GST-tagged proteins (150 μg) were simultaneously incubated with GSH-beads for 1 hr at 4°C on a rotating wheel. Beads were washed three times with buffer (150 mM NaCl, 50 mM HEPES/NaOH, pH 7.4, 1.5 mM $MgCl_2$, 0.1% NP-40), and the GSH-bound proteins were then incubated with Mon1-Ccz1 (25 μg) for 2 hr at 4°C on a rotating wheel. Beads were again washed three times in buffer. Bound proteins were eluted by boiling in SDS-sample buffer, resolved on SDS gels, and either analyzed by Coomassie Blue staining or immunoblotting with anti-CbP antibodies (*Lürick et al., 2017*).

## Light microscopy and image analysis

Yeast cells were first cultured in YPD media to log phase, and then switched to synthetic minimal medium lacking nitrogen (SD-N) for the indicated times to induce starvation. For CMAC staining of vacuoles, cells were incubated with 0.1 CMAC for 15 min at 30°C and subsequent washed with medium. Cells were imaged on a Deltavision Elite imaging system based on an inverted microscopy, equipped with 100x NA 1.49 and 60x NA 1.40 objectives, a sCMOS camera (PCO, Kelheim, Germany), an InsightSSI illumination system, and SoftWoRx software (Applied Precision, Issaquah, WA). Stacks of 6 or 8 images with 0.2 μm spacing were taken for constrained-iterative deconvolution (Soft-WoRx) and quantification.

## GEF assay on multilamellar vesicles (MLVs)

GEF assays were performed as described (*Nordmann et al., 2010*; *Cabrera et al., 2014*). 60 pmoles Atg8-His were incubated with 60 μl multilamellar vesicles (MLVs, 15 mM) of the following composition (palmitoyloleoyl phosphatidylcholine, 84 mol%, palmitoyloleoyl phosphatidylcholine 10 mol%, DOGS-NTA (1,2-dioleoyl-*sn*-glycero-3-[(N-(5-amino-1-carboxypentyl)iminodiacetic acid)succinyl]), 6 mol%) for 15 min at 25°C. 500 pmoles Ypt7-His were preloaded with MANT-GDP, and incubated with MLVs for 5 min at 25°C before addition of the Mon1–Ccz1 complex. MANT fluorescence was detected in a SpectraMax M3 Multi-Mode Microplate Reader (Molecular Devices, Germany). Samples were excited at 355 nm and fluorescence was detected at 448 nm. After 20–30 min, 0.1 mM GTP was added to trigger the exchange reaction. The decrease of MANT-GDP fluorescence is an indicator of nucleotide exchange.

## Giant Ape1 assay

Yeast cells (carry the plasmid pRS315-*CUP1pr-BFP-APE1*) were grown overnight in SDC-LEU medium, then diluted to early log phase next morning. 250 μM $CuSO_4$ was added to induce the giant Ape1 oligomer formation for 4 hr, and cultures were then switched to SD-N medium containing 250 μM $CuSO_4$ for 1 hr to induce autophagy.

## Acknowledgements

We thank Anna Lürick, Stephan Kiontke, and Claudio DeVirgilio for support and discussion, Sascha Martens and Ivan Dikic for constructs, and Kathrin Auffarth and Angela Perz for excellent technical assistance. DK is supported by the SFB944, Project P17. FR is supported by SNF Sinergia (CRSII3_154421), Marie Skłodowska-Curie ITN (765912), and ZonMW VICI (016.130.606) grants. JG received support by the SFB 944 graduate program. This work was funded by the DFG (UN111/7-3 and SFB 944, Project P11).

## Additional information

### Funding

| Funder | Grant reference number | Author |
|---|---|---|
| Deutsche Forschungsge-meinschaft | UN111/7-3 | Christian Ungermann |
| Deutsche Forschungsge-meinschaft | SFB 944 | Daniel Kuemmel Christian Ungermann |
| ZonMw | VICI 016.130.606 | Fulvio Reggiori |
| European Commission | Marie Skłodowska-Curie ITN | Fulvio Reggiori |
| Schweizerischer Nationalfonds zur Förderung der Wis-senschaftlichen Forschung | SNF Sinergia (CRSII3_154421 | Fulvio Reggiori |

The funders had no role in study design, data collection and interpretation, or the decision to submit the work for publication.

### Author contributions

Jieqiong Gao, Conceptualization, Data curation, Formal analysis, Methodology; Lars Langemeyer, Data curation, Formal analysis, Methodology; Daniel Kümmel, Conceptualization, Investigation; Fulvio Reggiori, Conceptualization, Funding acquisition, Investigation, Writing—review and editing; Christian Ungermann, Conceptualization, Data curation, Funding acquisition, Writing—original draft, Project administration

### Author ORCIDs

Christian Ungermann (iD) http://orcid.org/0000-0003-4331-8695

### Decision letter and Author response

Decision letter https://doi.org/10.7554/eLife.31145.032
Author response https://doi.org/10.7554/eLife.31145.033

## Additional files

### Supplementary files

• Supplementary file 1 Strains used in this study.
DOI: https://doi.org/10.7554/eLife.31145.026
• Supplementary file 2 Plasmids used in this study.
DOI: https://doi.org/10.7554/eLife.31145.027
• Transparent reporting form
DOI: https://doi.org/10.7554/eLife.31145.028

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
