## [Decision Letter]

Thank you for submitting your article "Molecular mechanism to target the endosomal Mon1-Ccz1 GEF complex to autophagosomes" for consideration by *eLife*. Your article has been reviewed by three peer reviewers, one of whom is a member of our Board of Reviewing Editors, and the evaluation has been overseen by Vivek Malhotra as the Senior Editor. The reviewers have opted to remain anonymous.

The reviewers have discussed the reviews with one another and the Reviewing Editor has drafted this decision to help you prepare a revised submission.

Summary:

This manuscript describes the mechanism how the Mon1-Ccz1 GEF complex localizes to autophagosomes. The authors show that Mon1-Ccz1 is recruited to autophagosomes by interaction with Atg8. Two putative LIR motifs are present in the C-terminal region of Ccz1. Mutagenic analysis of these LIR motifs suggests that the Ccz1-ATG8 interaction is important for autophagosome-lysosome fusion but not for endocytosis. Finally, using an in vitro GEF system, the authors show that recruitment of Mon1-Ccz1 by membrane-bound Atg8 can promote Ypt7 activation.

This study reveals a novel mechanism of recruitment of the Rab protein to autophagosomes and the data are mostly convincing. However, to fully support the authors' conclusion, the following issues need to be addressed.

Essential revisions:

1) The authors suggest that the endocytic pathway is not affected in Ccz1-LIR mutant cells at 37^o^C. However, this is not fully convincing from the data. In Figure 4, is the 2 h-long incubation at 37^o^C sufficient for degradation of preexisting CPY? Otherwise, it is difficult to detect a reduction in the amount of CPY after the temperature shift. What is the half-life of CPY? Also, to strictly follow the fate of Mup1 at 37^o^C in Figure 4, methionine should be added after the temperature shift. These pieces of information are critical to prove that the mutation in the LIR sequence of Ccz1 specifically impairs its interaction with Atg8 without affecting its general function.

2) Related to above criticism, the importance of the Atg8-Ccz1 interaction can also be tested by introducing mutations in the LIR-binding pocket in Atg8 (e.g., P52A/R67A mutation). Autophagosomes, if normally generated, should accumulate in these mutant cells.

3) Whether Ypt7 is present on autophagosomes or vacuoles (or both) is controversial. This is not carefully addressed in this study. The punctate structures on the vacuolar rim could represent either the PAS/ autophagosomes or a domain of the vacuolar membrane. In fact, in the review article by the authors (J Mol Biol (2017) 429:486), a "?" mark is added to Ypt7 on autophagosomes. Has this been already proved elsewhere? In not, the presence of Ypt7, Ccz1, and Mon1 on the autophagosomal side should be determined in more depth, for instance by biochemical methods (e.g., by purification of autophagosomes) or immunoelectron microscopy. It is also ideal to show that the Ccz1 complex is present on the outer membrane, not inside, of autophagosomes.

4) In Figure 2, normal colocalization of Ccz1 with Ape1 in *atg14* mutant cells is interesting and rather surprising. Is Atg8 also colocalized with Ape1 in the *atg14* mutant? Is there an Atg8-independent mechanism of Ccz1 targeting? In any case, the author should show actual images for Figure 2 (or in Supplemental Figures). It is also important to check the Ccz1-Ape1 colocalization in other atg mutants. Given the involvement of PI3K and potential link of Ccz1-Mon1/Ypt7 with endosomes, at least, atg2d, atg18d, and atg9d mutants should be added.

5) It is also important to test the possibility that Rab5 could be involved in the regulation of the PAS pool of Mon1-Ccz1 and Ypt7.

6) The interaction between Mon1-Ccz1 and Atg8 is not demonstrated in vivo. In particular, it is not clear whether this interaction is influenced by the lipidation status of Atg8. The authors should perform co-immunoprecipitation of endogenous proteins and pay attention to differentiate the two forms of Atg8.

[Editors' note: further revisions were requested prior to acceptance, as described below.]

Thank you for submitting your revised article "Molecular mechanism to target the endosomal Mon1-Ccz1 GEF complex to autophagosomes" for consideration by *eLife*. Your article has been reviewed by three peer reviewers, one of whom is a member of our Board of Reviewing Editors, and the evaluation has been overseen by Vivek Malhotra as the Senior Editor. The reviewers have opted to remain anonymous.

The reviewers have discussed the reviews with one another and the Reviewing Editor has drafted this decision to help you prepare a revised submission.

The revised manuscript has been substantially improved. Notably, the status of endocytic traffic and the role of PI3K are now clearer. However, despite the addition of these new data, this manuscript still contains some critical problems and the responses to the previous criticisms/comments are not sufficient.

1) The authors used the Atg8 I21R mutant instead of the P52A/R67A mutant to disrupt LIR-dependent interactions. However, characterization of the I21R mutant is missing. How does this mutation affect the substrate binding? Does it affect other functions of Atg8 besides substrate binding? Is there any previous study that used this mutant (if so, please cite it)? Furthermore, the authors did not determine whether autophagosomes accumulate in I21R mutant cells. This experiment is critical to rule out the possibility that Ccz1 LIR mutations affect other functions of the protein that are unrelated to Atg8-binding.

2) Whether Ypt7 is present on the autophagosomal membrane is one of the main issues of this study because the authors propose that Mon1-Ccz1 activates Ypt7 on autophagosomes. Additionally, in the Abstract, the authors state that "previous work implicated that endosomal Rab7/Ypt7 associates to autophagosomes prior to their fusion with lysosomes", but they do not specify which studies have suggested this. So far, the evidence that Ypt7 is on the autophagosomal membrane has been very limited. The authors show that Ypt7 colocalizes with Atg8 in vam3Δ cells, but it is unclear how they have ruled out the possibility that this represents tethering of an Atg8-positive autophagosome with Ypt7 on the vacuolar membrane. In the rebuttal letter, the authors claim that the amount of Ypt7 on the autophagosome is too small to be detected by immuno-EM. However, given that the fluorescent Ypt7 signals are clearly detected by IF (Figure 1), the authors could try immuno-EM a try. Alternatively, the authors may consider looking for large autophagosomes that can be clearly separated from the vacuole by immunofluorescence microscopy.

3) The authors' interpretation of Rab5-related data is self-contradictory. On one hand, they wanted to dismiss a role of Vps21 in autophagy (more on this later). On the other hand, they showed that Ccz1 dots were gone in vps21D. It is the authors' own claim that some Ccz1 dots are with Atg8, and acting to trigger Ypt7. With all Ccz1 dots gone (probably just too weak to be detected), one should actually expect defects in autophagy. While I acknowledge that subtle differences in strain background and experimental conditions might lead to some discrepancy, I'd be surprised that diminished recruitment of Ccz1 to autophagic membrane produces zero effect on autophagy (if so, what is the point of this manuscript?). The more rational interpretation is that vps21D only produces a partial kinetic defect (there are 3 genes in Rab5 family). In fact, the Cherry-Atg8 construct the authors used is not the ideal tool to assess partial defects (it functions substantially worse than GFP-Atg8, see Autophagy. 2015 Jun 3;11(6):954-60.) If the author really wants to dismiss vps21, they should at least use the quantitative Pho8D60 assay. My suggestion here is that they simply acknowledge that Vps21 (and by extension the Rab5 family) has a regulatory role in Ccz1 targeting, and revise their model and conclusions.

4) We suggested the authors to check the interaction of Ccz1 with Atg8 in vivo, and clarify whether Ccz1 preferentially interacts with the lipidated form of Atg8. It appears that the authors have completely missed the latter part. Demonstrating a stronger interaction (Figure 2) after starvation is totally irrelevant to the question as to whether lipidated Atg8 is the interactor. A potential technical issue is that the authors used GFP-Atg8 with a large tag, which makes it tricky (though not impossible) to discern the two forms. This can be resolved by using something like 3HA-Atg8. Researchers generally tend to believe that lipidated Atg8 is the critical factor in autophagy. It is likely the case here, even though the authors' in vitro experiment didn't directly address it either. That is why it is worth clarifying the interaction, especially considering that it is a very simple experiment. Imagine if the result turned out otherwise; the model would be quite different.

---

## [Author Response]

Essential revisions:1) The authors suggest that the endocytic pathway is not affected in Ccz1-LIR mutant cells at 37^o^C. However, this is not fully convincing from the data. In Figure 4, is the 2 h-long incubation at 37^o^C sufficient for degradation of preexisting CPY? Otherwise, it is difficult to detect a reduction in the amount of CPY after the temperature shift. What is the half-life of CPY? Also, to strictly follow the fate of Mup1 at 37^o^C in Figure 4, methionine should be added after the temperature shift. These pieces of information are critical to prove that the mutation in the LIR sequence of Ccz1 specifically impairs its interaction with Atg8 without affecting its general function.

We thank the reviewers for these important points. The half-life of CPY is 33.5 min. We took an additional control (the *vps11-1* temperature-sensitive strain, which impairs HOPS function in fusion at the vacuole) for the CPY assay to show that a 2h incubation at 37°C is sufficient to degrade preexisting CPY (Figure 4). Furthermore, to show that the LIR sequence of Ccz1 does not affect its general function in the endocytic pathway, we added methionine after the temperature shift. Indeed, the LIR sequence of Ccz1 exclusively impairs autophagy, but not its endolysosomal function (Figure 4). Likewise, vacuole morphology was comparable to wild-type under these conditions.

2) Related to above criticism, the importance of the Atg8-Ccz1 interaction can also be tested by introducing mutations in the LIR-binding pocket in Atg8 (e.g., P52A/R67A mutation). Autophagosomes, if normally generated, should accumulate in these mutant cells.

We agree with the reviewers that the interaction between Atg8 and Mon1-Ccz1 should be addressed in more detail. Therefore, we performed pull-down assays to test the interaction of Mon1-Ccz1 with two Atg8 mutants, the P52A R67A mutant and a second mutant (I21R), where we expected a direct impairment based on previous binding and structural studies. We now demonstrate that the interaction is weakly impaired by the P52A R67A mutant, but completely deficient in the Atg8 I21R mutant (Figure 2). As the I21R mutant shows that the positive charge now specifically impairs binding to Ccz1, in agreement with a direct LIR motif interaction, we included this result in the manuscript and show the double mutant, which based on structural considerations should only affect a subset of LIR motifs, here only in Author response image 1 for the reviewer.

**Author response image 1. respfig1:** Interaction of Atg8 mutants with Mon1-Ccz1. Purification of TAP-tagged Mon1-Ccz1 was

incubated with GST, GST-Atg8 and Atg8 mutants immobilized on GSH-Sepharose. Eluted proteins were resolved by SDS-PAGE and analyzed by Western blotting against the CbP-tag (top) or by Coomassie staining (bottom). Load, 5% (see Author response image 1).

3) Whether Ypt7 is present on autophagosomes or vacuoles (or both) is controversial. This is not carefully addressed in this study. The punctate structures on the vacuolar rim could represent either the PAS/ autophagosomes or a domain of the vacuolar membrane. In fact, in the review article by the authors (J Mol Biol (2017) 429:486), a "?" mark is added to Ypt7 on autophagosomes. Has this been already proved elsewhere? In not, the presence of Ypt7, Ccz1, and Mon1 on the autophagosomal side should be determined in more depth, for instance by biochemical methods (e.g., by purification of autophagosomes) or immunoelectron microscopy. It is also ideal to show that the Ccz1 complex is present on the outer membrane, not inside, of autophagosomes.

We agree with the reviewers that the localization of Ypt7 to autophagosomes is not demonstrated biochemically, but only by colocalization experiments. Yet our combined data along this line support the notion of a Mon1-Ccz1-dependent colocalization of Ypt7 with autophagic protein markers when autophagosome fusion with the vacuole is blocked (Figure 2). The pool of endogenous Ypt7 on autophagosomes is too low to be detected by immune EM-based techniques. For the endocytic pathway, we had to massively overproduce Ypt7 to find it on endosomes (Hönscher et al., 2014).

We are currently working on a parallel study, where we started to address the fusion of autophagosomes with vacuoles. In this context, we established a protocol for autophagosome purification. Within this study, we conducted a proteinase K protection assay of purified autophagosomes and could show that both Mon1-Ccz1 and Ypt7 are present on the outer membrane of autophagosomes. We present the data in this response letter to the reviewer, (Author response image 2) yet would like to present them in the context of our next study. We hope that the reviewers will agree with this. Nonetheless, we have added a sentence to the manuscript where we refer to this finding.

**Author response image 2. respfig2:** Biochemical Method for Obtaining an Autophagosome-enriched Fraction. (**A**) Scheme of the purification of autophagosomes from yeast. (**B**) Total cell lysates from cells grown in YPD medium and starved in SD-N medium for 3 hours. The 15,000 *g* pellet (P15) fraction from *vam3Δ* (Atg9-3xFlag, GFP-Atg8) were subjected to density gradient centrifugation and incubated with flag beads to pull down autophagosomes. (**C**) Detection of Mon1-Ccz1 and Ypt7 on autophagosomes by proteinase K-protection assay. Autophagosomes were collected as described in part B. Equal fractions were then treated with 1 mg/ml proteinase K (PK) in the absence or presence of 1% Triton X-100 (TX).

4) In Figure 2, normal colocalization of Ccz1 with Ape1 in atg14 mutant cells is interesting and rather surprising. Is Atg8 also colocalized with Ape1 in the atg14 mutant? Is there an Atg8-independent mechanism of Ccz1 targeting? In any case, the author should show actual images for Figure 2 (or in Supplemental Figures). It is also important to check the Ccz1-Ape1 colocalization in other atg mutants. Given the involvement of PI3K and potential link of Ccz1-Mon1/Ypt7 with endosomes, at least, atg2d, atg18d, and atg9d mutants should be added.

Indeed, we were also surprised that the loss of PI-3-P due to the *atg14Δ* deletion on autophagosomes did not affect localization of Ccz1. In response to the reviewers’ request, we tested for colocalization of Atg8 and Ape1 in the *atg14Δ* mutant and we could indeed confirm that these two proteins colocalize, probably at the PAS in agreement with previous studies showing that Atg8 is lipidated and present at the PAS in the absence of Atg14 (Suzuki et al., 2001; 2007). Thus, the pool of Atg8 present at the PAS is sufficient for Mon1-Ccz1 recruitment to autophagosomal membranes.

We also generated *atg2*Δ, *atg18*Δ and *atg9*Δ mutants and examined colocalization of Ccz1 and Ape1 (Figure 2). Finally, we placed the actual images for the bar graphs of Figure 2 in new Supplemental Figures.

5) It is also important to test the possibility that Rab5 could be involved in the regulation of the PAS pool of Mon1-Ccz1 and Ypt7.

We tested whether Vps21 is required for autophagy by fluorescence microscopy (Figure 1—figure supplement 2). Our data demonstrate that the Rab5-like Vps21 does not impair autophagy under nitrogen starvation. As we are not sure if this is due to the selected background strain, we conducted the assay with another strain (SEY6210), though observed the same result. In both strains, Ccz1 was found primarily in the cytosol, yet autophagy was still functional as observed by mCherry-Atg8 in the vacuole lumen. We thus believe that the role of Vps21 is restricted to the endocytic pathway as overproduced Vps21 and Vps8, which results in an accumulation of endosomes proximal to vacuoles (Markgraf et al., 2009), also accumulates Mon1-Ccz1 at this site. However, these conditions do not redirect Mon1-Ccz1 exclusively to endosomes.

We do not yet understand the discrepancy to the recent study of Zhou et al. (Zhou et al., 2017), which suggested a Vps21 involvement in autophagy. We suspect that this effect could be due to the selected background strain, and itwill be important to dissect direct from indirect contributions. We like to add that a role of Vps proteins (except for the Vps proteins involved in fusion with vacuoles) in the autophagy pathway is controversial, as they were also not identified in the principal autophagy screens performed by the Ohsumi, Klionsky and Thumm laboratories.

6) The interaction between Mon1-Ccz1 and Atg8 is not demonstrated in vivo. In particular, it is not clear whether this interaction is influenced by the lipidation status of Atg8. The authors should perform co-immunoprecipitation of endogenous proteins and pay attention to differentiate the two forms of Atg8.

We have immunoprecipitated endogenously TAP-tagged Ccz1 from a strain expressing GFP-Atg8 to demonstrate the interaction of Mon1-Ccz1 and Atg8 in vivo. We indeed observed more Ccz1 in complex with Atg8 when cells were starved prior to lysis. This agrees with our model of starvation-induced relocalization of Mon1-Ccz1 to the autophagosomal surface.

[Editors' note: further revisions were requested prior to acceptance, as described below.]

The revised manuscript has been substantially improved. Notably, the status of endocytic traffic and the role of PI3K are now clearer. However, despite the addition of these new data, this manuscript still contains some critical problems and the responses to the previous criticisms/comments are not sufficient.

We thank the Reviewers for recognizing the novelty of our study and for their constructive comments regarding our revised manuscript. In response to their reassessment, we now:

1) Characterized Atg8 (I21R), which is impaired in selective autophagy (Figure 2—figure supplement 3).

2) Extended Figure 1—figure supplement 1 by the giant Ape1 assay to provide further evidence that Ypt7 is present on the autophagosomal membrane.

3) Provide evidence that the *vps21Δ* mutant indeed has an autophagy defect by using GFP-Atg8 instead of mCherry-Atg8, and by employing the Pho8Δ60 assay in Figure 1—figure supplement 2.

4) Demonstrate that Ccz1 mainly interacts with the lipidated Atg8 in vivoby immunoprecipitation of endogenous Mon1-Ccz1 with GFP-Atg8 from wild-type and *atg4Δ* cells (Figure 2).

1) The authors used the Atg8 I21R mutant instead of the P52A/R67A mutant to disrupt LIR-dependent interactions. However, characterization of the I21R mutant is missing. How does this mutation affect the substrate binding? Does it affect other functions of Atg8 besides substrate binding? Is there any previous study that used this mutant (if so, please cite it)? Furthermore, the authors did not determine whether autophagosomes accumulate in I21R mutant cells. This experiment is critical to rule out the possibility that Ccz1 LIR mutations affect other functions of the protein that are unrelated to Atg8-binding.

We have extended our analysis here using Atg8 I21R mutant. The I21R mutant has a defect in selective autophagy (Ape1 processing), though can support bulk autophagy at least in the background tested here (Pho8Δ60 assay) (Figure 2—figure supplement 3). In this sense, it behaves similar to the suggested P52A/P67A mutant of Atg8 (Noda et al., 2008; Okamoto et al., 2012, JBC), which again showed its strongest defect in selective autophagy, while supporting bulk autophagy.

We therefore believe that Mon1-Ccz1 recognizes Atg8 also via this site, but may employ additional binding sites in Atg8 for its targeting to autophagosomes during starvation. We would like to note that the diversion of Mon1-Ccz1 from endosomes to autophagosomes during starvation may also require additional posttranslational modifications, possibly even in the vicinity of the LIR motif.

The observed loss of Mon1-Ccz1 binding to Atg8 in the two Atg8 mutants may reflect the mode how Mon1-Ccz1 recognizes in part Cvt vesicles, while a possible phosphorylation of Ccz1 or Mon1 could make Mon1-Ccz1 available for Atg8 during starvation. This is certainly an issue that warrants future analysis.

2) Whether Ypt7 is present on the autophagosomal membrane is one of the main issues of this study because the authors propose that Mon1-Ccz1 activates Ypt7 on autophagosomes. Additionally, in the Abstract, the authors state that "previous work implicated that endosomal Rab7/Ypt7 associates to autophagosomes prior to their fusion with lysosomes", but they do not specify which studies have suggested this. So far, the evidence that Ypt7 is on the autophagosomal membrane has been very limited. The authors show that Ypt7 colocalizes with Atg8 in vam3Δ cells, but it is unclear how they have ruled out the possibility that this represents tethering of an Atg8-positive autophagosome with Ypt7 on the vacuolar membrane. In the rebuttal letter, the authors claim that the amount of Ypt7 on the autophagosome is too small to be detected by immuno-EM. However, given that the fluorescent Ypt7 signals are clearly detected by IF (Figure 1), the authors could try immuno-EM a try. Alternatively, the authors may consider looking for large autophagosomes that can be clearly separated from the vacuole by immunofluorescence microscopy.

We have responded to this criticism in our previous submission and provided the reviewer with evidence of purified autophagosomes contain Ypt7 on their surface. Moreover, Hegedus et al., 2016 showed Rab7 binding to autophagosomes in *Drosophila*. We thus felt that we had addressed their concern, yet recognize that additional support would be needed.

It is tempting to believe that a fluorescent signal is sufficient to also localize a protein by immuno-electron microscopy (IEM). However, the Reggiori group has long experience with yeast and requires at least 1900 molecules/cell to reliably recover an IEM signal. As Ypt7 is not as abundant if not overexpressed, we turned to the giant Ape1 assay to enrich a possible immature structure to visualize Ypt7 by fluorescence microscopy. We conducted this assay both under fusion compromised conditions in the *vam3*Δ mutant, which results in a massive vacuole fragmentation and loss of autophagosome-vacuole contact, and in wild-type. Our data now provide evidence that Ypt7 is found on the cup-shaped isolation membrane as dots in wild-type and *vam3*Δ background. These data have been added now as Figure 1—figure supplement 1 to the manuscript.

Moreover, Yamano et al., 2018 (*eLife*) just recently showed that MON1-CCZ1 is required for the mitochondrial recruitment of RAB7A during mitophagy in mammalian cultured cells.

3) The authors' interpretation of Rab5-related data is self-contradictory. On one hand, they wanted to dismiss a role of Vps21 in autophagy (more on this later). On the other hand, they showed that Ccz1 dots were gone in vps21D. It is the authors' own claim that some Ccz1 dots are with Atg8, and acting to trigger Ypt7. With all Ccz1 dots gone (probably just too weak to be detected), one should actually expect defects in autophagy. While I acknowledge that subtle differences in strain background and experimental conditions might lead to some discrepancy, I'd be surprised that diminished recruitment of Ccz1 to autophagic membrane produces zero effect on autophagy (if so, what is the point of this manuscript?). The more rational interpretation is that vps21D only produces a partial kinetic defect (there are 3 genes in Rab5 family). In fact, the Cherry-Atg8 construct the authors used is not the ideal tool to assess partial defects (it functions substantially worse than GFP-Atg8, see Autophagy. 2015 Jun 3;11(6):954-60.) If the author really wants to dismiss vps21, they should at least use the quantitative Pho8D60 assay. My suggestion here is that they simply acknowledge that Vps21 (and by extension the Rab5 family) has a regulatory role in Ccz1 targeting, and revise their model and conclusions.

We have conducted multiple assays to address the role of Vps21 in autophagy, and find as published reduced bulk autophagy in the *vps21Δ* mutant using the Pho8Δ60 assay. We also observe that Ccz1 poorly localizes to membranes in the *vps21Δ* mutant and does not concentrate upon starvation. We do not yet know if the loss of Vps21 also impairs Ccz1 targeting to autophagosomes as a high cytosolic signal might not allow us to see the autophagosomal pool. We now have changed the text to adjust our statements here.

4) We suggested the authors to check the interaction of Ccz1 with Atg8 in vivo, and clarify whether Ccz1 preferentially interacts with the lipidated form of Atg8. It appears that the authors have completely missed the latter part. Demonstrating a stronger interaction (Figure 2) after starvation is totally irrelevant to the question as to whether lipidated Atg8 is the interactor. A potential technical issue is that the authors used GFP-Atg8 with a large tag, which makes it tricky (though not impossible) to discern the two forms. This can be resolved by using something like 3HA-Atg8. Researchers generally tend to believe that lipidated Atg8 is the critical factor in autophagy. It is likely the case here, even though the authors' in vitro experiment didn't directly address it either. That is why it is worth clarifying the interaction, especially considering that it is a very simple experiment. Imagine if the result turned out otherwise; the model would be quite different.

In response to the reviewers’ comment, we have used the *atg4Δ* deletion and monitored Mon1-Ccz1 association with Atg8 by pull-down. Without Atg4, Atg8 is not targeted to the autophagosome, even though it is present in cells. In agreement with our interpretation, we only find a greatly enhanced interaction of Ccz1 and Atg8 upon starvation in wild-type cells, whereas this interaction was lost in *atg4Δ* deletion cells (Figure 2), which also correlates to our microscopy data that Ccz1 fails to localize to the PAS in all the mutants blocking Atg8 conjugation to PE (Figure 2; Figure 2—figure supplement 1). Therefore, we conclude that Ccz1 mainly recognize lipidated Atg8 on the autophagic structures.